# Recombinant Glycoprotein E of Varicella Zoster Virus Contains Glycan-Peptide Motifs That Modulate B Cell Epitopes into Discrete Immunological Signatures

**DOI:** 10.3390/ijms20040954

**Published:** 2019-02-22

**Authors:** Rickard Nordén, Jonas Nilsson, Ebba Samuelsson, Christian Risinger, Carina Sihlbom, Ola Blixt, Göran Larson, Sigvard Olofsson, Tomas Bergström

**Affiliations:** 1Department of Infectious Diseases, Institute of Biomedicine, Sahlgrenska Academy at the University of Gothenburg, 41346 Gothenburg, Sweden; ebba.samuelsson@gu.se (E.S.); sigvard.olofsson@microbio.gu.se (S.O.); tomas.bergstrom@microbio.gu.se (T.B.); 2Department of Clinical Chemistry and Transfusion medicine, Institute of Biomedicine, Sahlgrenska Academy at the University of Gothenburg, 41346 Gothenburg, Sweden; jonas.nilsson@clinchem.gu.se (J.N.); goran.larson@clinchem.gu.se (G.L.); 3Chemical Glyco-Biology section, Department of Chemistry, University of Copenhagen, 1871 Copenhagen, Denmark; christian.risinger@gmail.com (C.R.); olablixt@chem.ku.dk (O.B.); 4Proteomics Core Facility, Sahlgrenska Academy at the University of Gothenburg, 41346 Gothenburg, Sweden; carina.sihlbom@gu.se

**Keywords:** antibody binding, B cell epitope, glycoprotein E, glycosylation, vaccine, varicella zoster virus

## Abstract

A recombinant subunit vaccine (Shingrix^®^) was recently licensed for use against herpes zoster. This vaccine is based on glycoprotein E (gE) of varicella zoster virus (VZV), the most abundantly expressed protein of VZV, harboring sites for N- and O-linked glycosylation. The subunit vaccine elicits stronger virus-specific CD4+ T cell response as well as antibody B cell response to gE, compared to the currently used live attenuated vaccine (Zostavax^®^). This situation is at variance with the current notion since a live vaccine, causing an active virus infection, should be far more efficient than a subunit vaccine based on only one single viral glycoprotein. We previously found gE to be heavily glycosylated, not least by numerous clustered O-linked glycans, when it was produced in human fibroblasts. However, in contrast to Zostavax^®^, which is produced in fibroblasts, the recombinant gE of Shingrix^®^ is expressed in Chinese hamster ovary (CHO) cells. Hence, the glycan occupancy and glycan structures of gE may differ considerably between the two vaccine types. Here, we aimed at (i) defining the glycan structures and positions of recombinant gE and (ii) identifying possible features of the recombinant gE O-glycosylation pattern contributing to the vaccine efficacy of Shingrix^®^. Firstly, recombinant gE produced in CHO cells (“Shingrix situation”) is more scarcely decorated by O-linked glycans than gE from human fibroblasts (“Zostavax situation”), with respect to glycan site occupancy. Secondly, screening of immunodominant B cell epitopes of gE, using a synthetic peptide library against serum samples from VZV-seropositive individuals, revealed that the O-linked glycan signature promoted binding of IgG antibodies via a decreased number of interfering O-linked glycans, but also via specific O-linked glycans enhancing antibody binding. These findings may, in part, explain the higher protective efficacy of Shingrix^®^, and can also be of relevance for development of subunit vaccines to other enveloped viruses.

## 1. Introduction

Herpes zoster (HZ) or shingles is a result of reactivation of latent varicella zoster virus (VZV) from dorsal root or cranial nerve ganglia. The seroprevalence of VZV IgG exceeds 90% in most adult populations and, thereby, a large number of individuals are at risk of developing HZ. A live attenuated vaccine (Zostavax^®^, Merck) is licensed for use in individuals ≥50 years of age. However, the protective efficacy against HZ is rather modest, at 51.3%, and declines with increasing age, i.e., 66.5% between 50 and 59 years, and only 37.6% above the age of 70 [1,2]. Recently, a recombinant subunit vaccine (HZ/su, Shingrix^®^, Glaxo Smith-Kline Biologicals) was developed and licensed for use against HZ. This vaccine is based on the single glycoprotein E (gE) and the adjuvant AS01_B_. This envelope protein is highly expressed on VZV viral particles and on infected cells, and functions as a Fc receptor for human immunoglobulin G (IgG) [3]. The efficacy of the subunit vaccine is as high as 97.2%, and with no sign of declining potency with increasing patient age [4,5]. Earlier clinical trials have shown that one dose of the recombinant subunit vaccine elicits stronger gE and VZV-specific CD4+ T cell and antibody B cell responses than two doses of the live attenuated vaccine [6,7,8]. When both vaccines were administered simultaneously, there was no increase either in cellular or humoral responses to gE, as compared to giving the recombinant vaccine alone [6]. Recent comparative studies have highlighted that the superior efficacy of the recombinant subunit vaccine can be explained by its ability to generate a higher T cell memory response compared to the live vaccine [9,10]. Still, the question remains: How can a vaccine based only on one single glycoprotein outcompete a live virus (although attenuated) vaccine that actually results in an active virus infection?

Human herpesviruses belong to the *Herpesviridae* family of large enveloped DNA viruses, where the genomes have the capacity to encode between ~70 and ~230 proteins, including various glycoproteins that protrude from the viral surface [11]. Many of these proteins are engaged in subversion of the host immune system, a critical prerequisite for human herpesviruses to establish a lifelong latent infection in their host, which is a hallmark for these viruses [12]. A prevailing notion in vaccinology is that live attenuated VZV virus mimics an authentic infection, thereby mounting a complete B and T cell response with capacity to address all of the almost 100 proteins introduced by VZV [13,14]. This includes all components of the viral particle, especially all glycoproteins that are present in the viral particle, but also exposed on the surface of infected cells. Yet, the subunit vaccine, which is based solely on one single viral glycoprotein designated gE, can provide a B and T cell response against this protein that even surpasses that of the live attenuated vaccine [1,6,7].

The high potency of gE as a vaccine immunogen, irrespective if it is administered as a subunit or live HZ vaccine, probably depends on several factors, not least its predominance at the surface of VZV viral particles. However, another important aspect of gE is its structural organization; it harbors two consensus N-glycosylation sites (Asn–X–Ser/Thr) but also contains a so-called mucin-like domain (MLD), a peptide stretch that is rich in Pro, Ser, and Thr residues, and constitutes a target for mucin-type O-linked glycosylations [15,16]. O-linked glycosylation of viral MLDs is initiated by addition of *N*-acetylgalactosamine (GalNAc) via an O-glycosidic linkage to single or combinations of multiple Ser and Thr residues of the MLD [17,18]. Owing to viral interference with O-glycan elongation, much of these carbohydrate components of viral MLDs are composed of shorter glycans, including O-linked GalNAc monosaccharides (Tn antigen) and GalGalNAc disaccharides (T antigen) [16,19,20]. Recently, we established that such truncated O-linked glycans, in combination with the proximate peptide sequence at the glycosylation site, forms a novel variant of viral B cell epitope to which prominent antibody responses are raised in the infected patient [21,22,23]. Importantly, it was shown that a single GalNAc unit and each specific amino acid of the glycopeptide stretch contributed equally well to the B cell epitope specificity and avidity towards the cognate antibody [22,23]. Although the exact contribution of the MLD to the VZV gE subunit vaccine is not clarified, it is interesting to note that another successful subunit viral vaccine, i.e., the hepatitis B virus (HBV) subunit vaccine, is also based on a single viral envelope glycoprotein containing multiple MLD stretches [24,25]. Moreover, we recently defined the attachment sites of O-linked glycans within VZV gE obtained from infected fibroblasts and from a clinical sample, including several sites within the MLD domain of the protein [16].

The contribution of viral MLDs to glycopeptide B cell epitope is still incompletely understood. This is at least partly due to the complexity of both initiation and elongation of O-linked glycosylation, which confers a natural variability in the expressed structures. Thus, firstly, O-linked glycosylation may or may not occur at a specific amino acid residue and, secondly, different glycan structures may be found at that same position, depending on expression levels of glycosyltransferases which vary between cell and tissue types, but also between different species [26,27,28]. Hence, the choice of the cell system used for vaccine production is a critical parameter to ensure production of an optimal immunogen with respect to presentation of adequate T and B cell glycopeptide epitopes.

We have previously defined, in detail, the GalNAc occupancy of the potential O-glycosylation sites of human fibroblast-produced gE, the same cell type that is used for production of the live attenuated HZ vaccine [16]. In the present study, we defined the site-specific glycan structures of the recombinant gE expressed in Chinese hamster ovary (CHO-K1) cells, the same cell line used to produce the subunit vaccine, and show (i) that the glycans are important for antibody recognition and (ii) that there are distinct differences in the O-glycan occupancy compared with gE derived from human fibroblasts. In addition, we aimed at defining the immunodominant B cell epitopes of the recombinant gE using synthetic peptides and glycopeptides by screening of serum samples from VZV-seropositive and VZV-seronegative individuals. 

## 2. Results

### 2.1. Glycosylation of the Recombinant gE Produced in CHO-K1 Cells

The live HZ vaccine and the subunit HZ vaccine are derived from different types of producer cell lines, which is likely to result in important differences in the glycoprotein glycosylation status between the two vaccine constructs. VZV gE, produced in human diploid fibroblasts as part of whole virus infection and used for Zostavax^®^ live virus vaccine production, has been characterized in detail with respect to the O-linked glycosylation of this protein [16]. Here, we first aimed at structurally characterizing the O-linked glycans and glycosylation positions of a VZV gE construct (gE HZ/su prototype (gE_HP_)) derived from CHO-K1 cells, the cell line used for production of the Shingrix^®^ subunit vaccine (HZ/su), to reveal possible differences in O-linked glycosylation of gE between the two vaccine variants. To distinguish between the two variants, gE derived from VZV infected human diploid fibroblasts is hereafter referred to gE live (gE_L_). 

In order to define the site-specific glycan structures of the gE_HP_, the isolated protein was subjected to liquid chromatography tandem mass spectrometry (LC–MS/MS) analysis. At first, gE_HP_ was subjected to on-membrane trypsin cleavage. This enabled only 34% coverage of the total protein length when the tryptic peptides and glycopeptides were analyzed with LC–MS/MS. In order to also study peptide stretches that may be suboptimal for tryptic cleavage sites in gE_HP_, an additional on-membrane pronase treatment was included in order to generate additional peptide fragments by increasing the number of potential cleavage sites. This resulted in an increased coverage, with a total coverage of 62% when the two methods were combined (Appendix A). We used ETD-induced fragmentation of the trypsin-digested glycopeptides to provide peptide backbone fragmentation into the c- and z-ion series while keeping the glycans intact, thus pinpointing glycosylation sites (Appendix A). Also, HCD-induced glycosidic fragmentation of the glycans and simultaneous peptide fragmentation into the b- and y-ion series was undertaken to identify additional O-glycopeptides, and to define the structures of individual glycans (Figure 1 and Appendix A). In total, we found eleven Ser and Thr residues to be glycosylated, which were found in three clusters of gE_HP_, as shown in Figure 2 and Appendix A. The O-linked glycans were all composed of the core 1 HexHexNAc structure (Galβ3GalNAcα1-*O*-) and were mainly disialylated with *N*-acetylneuraminic acid (NeuAc) both at the inner GalNAc and at the terminal Gal (Figure 1A,B and Figure 2) in the O-glycan epitope cluster I (GC-I). Monosialylated variants of the core 1 type dominated in the GC-II and GC-III clusters (Figure 1A,B and Figure 2).

Also, an additional O-acetyl group of NeuAc (NeuAc(Ac)) was identified both in the GC-I and in the GC-II cluster (Figure 1C, Figure 2 and Appendix A). It was signified by the presence of diagnostic oxonium ions corresponding to NeuAc(Ac) and NeuAc(Ac)–H_2_O at *m/z* 334.11 and *m/z* 316.10 (Figure 1C), whereas the corresponding NeuAc oxonium ions only possess the *m/z* 292.10 and 274.09 ions (Figure 1A,B). Moreover, the nonhuman sialic acid *N*-glycolylneuraminic acid (NeuGc) was also detected in the GC-I and the GC-II cluster (Figure 1D, Figure 2 and Appendix A). The NeuGc-specific oxonium ion *m/z* 308.10 and 290.09 were here used for the diagnostic purposes. When the *m/z* 274.09, *m/z* 316.10, and *m/z* 290.09 ions were screened for in extracted ion chromatograms, it was evident that the relative abundance of the NeuAc(Ac)- and NeuGc-containing glycoforms were approximately at the 1%–2% level compared to that of NeuAc. There were observable differences in attachment site occupancy found for different peptides of the GC-I and GC-III clusters (Figure 2 and Appendix A). However, both non-glycosylated and glycosylated peptides were only found simultaneously for two of the trypsin-digested regions (marked *None* regarding modification in Appendix A). For the 73-KAYDHNSPYIWPR-85 region, 3%–16% of the peptides were found to be glycosylated, assuming that the glycopeptides and peptides ionize equally well (Appendix A). In comparison, for the 524-EITPVNPGTSPLLR-537 peptides, the degree of glycosylation was 54% (Appendix A). 

There are two potential sites for N-linked glycosylation within gE_HP_ protein sequence (Asn-266 and Asn-437). The peptide coverage enabled analysis only of Asn-437, which was indeed found to carry an N-linked glycan (Figure 2 and Appendix A). The N-linked glycan was of the complex type with 2–3 antennae each carrying an *N*-acetylglucosamine (GlcNAc) followed by a galactose (Gal) residue, and up to two NeuAc residues on the triantennary structures (Appendix A). The inner GlcNAc residue attached to the asparagine Asn-437 also carried an additional fucose (Fuc) residue. No NeuGc- or NeuAc(Ac)-containing glycoforms could be detected for the N-glycopeptides.

### 2.2. Removal of Glycans Negatively Impact Recognition of gE_HP_ by Polyclonal Human Sera

The full length gE_HP_ reacted strongly with polyclonal sera isolated from individuals with antibodies against VZV, as previously observed [31]. In order to assess the impact of the glycan structures on the serological response, we also trimmed and enzymatically removed the glycans on gE_HP_ prior to ELISA. The different enzymatic treatments conferred distinct size shifts as determined by gel electrophoresis, indicating complete removal of glycans (Figure 3A). There was reduced reactivity towards human polyclonal sera from VZV-positive individuals with high or intermediated antibody titers (high titer n = 5, intermediate titer n = 5) when the deglycosylated gE_HP_ was analyzed by ELISA (Figure 3B). Removal of N-linked glycans by PNGaseF treatment and O-linked glycans by O-glycosidase treatment independently reduced the reactivity by approximately 8% each, compared to the mock-treated control, whereas removal of only the sialic acids did not confer any significant change in serum reactivity. Removal of both N- and O-linked glycans did seem to have an additive effect since there was a 17% reduction in reactivity in this sample compared to mock-treated. The untreated control sample, that was not subjected to any treatment but added at a higher concentration, behaved as the mock-treated control sample (Figure 3B). The enzymatic hydrolysis for removal of O-linked and N-linked glycans was almost complete as indicated by the distinct size shifts in the gel. Hence, the majority of the reactivity of VZV-positive sera towards gE_HP_ is restricted to peptide epitopes, despite the 17% contribution of reactivity that was obviously positively modulated by glycans (Figure 3B). However, this figure is the net effect of carbohydrate modulation of seroreactivity that may also include peptide regions of gE_HP_ for which glycosylation interferes with the seroreactivity of particular epitopes of the protein.

### 2.3. Single GalNAc Moieties Attached to Peptides Modulate Seroreactivity of Discrete gE_HP_ Epitopes

To define single discrete epitopes and their relationship to modulation by O-linked glycans of the gE_HP_, we generated a library of 16-, 18-, and 20-mer synthetic peptides covering 49% of the recombinant protein (Appendix A) using previously published methods [22]. The peptides were produced to cover the domains of gE_HP_ that were previously shown to be immunodominant [29] and also to cover the domains that carry glycan modifications as determined by LC–MS/MS (Figure 2 and Appendix A). The peptides were unglycosylated or carried single O-linked GalNAc modifications on serine, threonine, or tyrosine residues. The peptides and glycopeptides were immobilized on a microarray glass slide, and thereafter incubated with serum from verified VZV-positive (individual serum samples n = 11 and pooled serum samples n = 2). The microarray slides were scanned, and the mean intensities of fluorescence were determined, and average relative fluorescence unit (RFU) datasets were generated from all serum samples (Appendix A). The number of different VZV-positive serum samples (IgG) that reacted to the epitopes ranged between 0 and 10 (Appendix A). We found that the antibody (glyco)peptide epitopes of gE_HP_, reactive with the VZV-positive serum samples, were confined to four distinct clusters along the protein amino acid sequence. The details are presented in Figure 2 and Appendix A. When all the peptides and glycopeptides were analyzed, three patterns emerged: (i) addition of a GalNAc moiety conferred a reduction in average RFU-value, i.e., interference, (ii) addition of a GalNAc moiety had no effect, and (iii) addition of a GalNAc moiety conferred an increase in average RFU-value, i.e., enhancement. Interference of antibody binding was most prominently observed within the region spanning W64-G140, including GC-I (Figure 2 and Appendix A), the domain previously described as the most antigenic [29]. We found a significant reduction in reactivity when a single GalNAc residue was added either to a tyrosine or to a serine (Tyr-75, Ser-79, Tyr-81, and Tyr-88) of peptide 73-KAYDHNSPYIWPRNDYDG-90 (Figure 4A). Throughout the protein sequence, we could observe peptide epitopes where addition of a GalNAc residue did not affect antibody binding, but this situation was most evident in the region between GC-I and GC-II (Figure 2 and Appendix A). Enhancement of antibody binding was observed for peptide 460-GTTLKFVDTPESLSGLYV-477 when a GalNAc residue was added to Ser-473 (Figure 4B). 

### 2.4. Detailed Characterization of Reactivity against Synthetic Peptides for Individual Sera

The individual serum RFU-values, against synthetic peptides either unglycosylated or carrying single O-GalNAc modifications at the serine, threonine, or tyrosine residues, were further characterized in detail. We examined the RFU-values for each peptide and each individual serum sample (n = 11) in order to pinpoint the effect of glycosylation. When examining the previously described immunodominant epitope spanning W64-G90, we found that peptide 64-WVNRGESSRKAYDHNSPY-81 was low- or non-reactive to all individual sera, both when unglycosylated and when glycosylated at the potential sites for O-glycosylation (Figure 5A). On the contrary, peptide 73-KAYDHNSPYIWPRNDYDG-90 was highly reactive in its unglycosylated state, with all but one individual serum sample reacting to the peptide and displaying high RFU-values, and when single GalNAc residues were added to the peptide, the reactivity was abolished or severely reduced in all cases (Figure 5B). A closer inspection of the G111–G140 domain revealed a pattern where only a more limited stretch of the epitope showed reactivity. The unglycosylated peptide 111-GERLMQPTQMSAQEDLGDDT-130 was non-reactive, and addition of one GalNAc did not significantly alter the response (Figure 5C). The sequential peptide, i.e., 121-SAQEDLGDDTGIHVIPTLNG-140, was evidently more reactive with three individual sera displaying high RFU-values to the unglycosylated peptide. However, addition of a single GalNAc residue to Ser-121 and Thr-137 appeared to enhance the RFU-values when the same serum samples were added to this peptide stretch (Figure 5D). It thus appears that the B cell epitopes are narrow peptide stretches that confer reactivity to individual sera, and that addition of a glycan epitope can either decrease or increase the observed RFU-values in a uniform pattern, i.e., serum from the same individual either show increased or decreased reactivity depending on the addition of a GalNAc moiety.

When analyzing peptides covering the stem region of the protein including the MLD, which harbors several O-linked glycans, an even more heterogeneous pattern emerged. Peptide 460-GTTLKFVDTPESLSGLYV-477 showed no reactivity to any of the individual serum samples in its unglycosylated state, whereas four individual sera showed enhanced, albeit modest, RFU-values when the peptide carried a GalNAc residue at Ser-473 (Figure 6A). The stem region closer to the transmembrane domain showed a more distinct effect of glycosylation on the serum response. Peptides covering E505–A540 showed modest reactivity to only a few individual sera when they were unglycosylated (Figure 6B–D). However, when a GalNAc residue was added to either Thr-512, Thr-519, or Thr-520, there was a marked increase in RFU-values from serum samples 1, 2, and 7 (Figure 6B). Addition of a GalNAc residue to Thr-519, Thr-520, or Thr-526 increased RFU-values from serum 4, 6, and 11 (Figure 6C) and addition of a GalNAc residue to Thr-526 or Ser-533 conferred increased RFU-values from serum 1, 6, 7, 9, and 11 (Figure 6D). In the MLD domain of the stem region, individual serum samples thus exhibited altered reactivity depending on where the GalNAc residue was introduced within the peptide sequence, generating discrete immunological profiles for the respective individual serum sample. 

## 3. Discussion

Although direct head-to-head comparative clinical studies of immune protection against HZ currently are lacking, the newly approved subunit vaccine Shingrix^®^ (HZ/su), based on gE of VZV, demonstrates efficacy data that seem to be superior to those of Zostavax^®^, the currently used live attenuated vaccine [1,4,5]. At a first glance, this phenomenon may appear paradoxical, as live vaccines are considered to be optimal immunogens, as they mimic the original virus infection and activate all branches of the acquired immune response to target virtually all proteins encoded by the viral genome [32]. By contrast, a subunit vaccine is able to induce acquired immunity to only one or a few viral proteins contained in the vaccine, and it often depends on the quality of added adjuvants to compensate for the strong B and T cell responses induced by active viral replication. However, the status of most viral glycoproteins, including VZV gE, as genuine membrane-spanning proteins, makes them unique targets for two important antiviral effects of the B cell response. First, as they protrude from the viral envelope, they function as targets for neutralizing antibodies and, second, owing to their position as constituents in the virus-infected cell, they serve as targets for antibodies, participating not only in antibody-dependent cellular cytotoxicity (ADCC) but also in prevention of viral cell-to-cell spread. However, the present knowledge regarding B cell responses to gE, following vaccination with any of the two HZ vaccines, is sparse. It has been reported that Zostavax^®^ induces antibodies, primarily against gE, as determined by clonal expansion of B cells [33], but no such data are available for the recombinant subunit vaccine, although it elicits a stronger gE-specific Th1 memory response compared to the live attenuated vaccine [9,10].

Since the immunocompetent individual who has encountered a primary VZV infection is believed to be protected from disease caused by reinfection, knowledge on the different antibody specificities generated by primary VZV infection is also of relevance for understanding of the function of a successful HZ vaccine. Here, we compared the reactivity to VZV gE of naturally occurring antibodies in sera drawn from patients, with special reference to possible differences in O-linked glycosylation of the gE species of the subunit and live vaccine formulations, respectively. This strategy was based on the following considerations: (i) The two most efficient subunit vaccines to enveloped viruses, i.e., VZV gE against HZ and HBsAg against HBV infection, are both based on viral glycoproteins containing MLDs decorated by numerous short O-linked glycans [15,16,24,25]. (ii) Short O-linked glycans, together with their surrounding oligopeptide environment, constitute immunoreactive epitopes in viral glycoproteins with MLD [21,34]. (iii) In contrast to N-linked glycans being added by one single enzyme species, strongly homologous over the entire animal kingdom, the initiation of O-linked glycans is carried out by twenty different isotypes of GalNAc transferases, differing in target specificity [27,35]. This results in prominent, species- and tissue-dependent differences in the glycan occupancy patterns of the potential O-glycosylation sites of one and the same viral glycoprotein MLD. (iv) Shingrix^®^ is derived from a hamster ovary cell line, whereas Zostavax^®^ is derived from human fibroblasts, providing potential for considerable differences in the O-glycan occupancy pattern of gE between the two immunogens [36]. 

When analyzing the global O-glycosylation pattern, the most prominent difference between gE_HP_ as determined in the present study and previously published data for gE_L_ is that gE_L_ is considerably more densely O-glycosylated than gE_HP_, i.e., a considerably higher number of the potential O-glycosylation sites are occupied by glycans [16]. This difference was, in particular, dominant for O-glycan epitope cluster I (GC-I; Figure 2), harboring epitopes reacting with a majority of sera from VZV-infected patients. The present data, based on analysis of glycosylated and non-glycosylated peptides representing the area containing the linear B cell epitope (including GC-I), clearly demonstrated that O-glycosylation interfered with the reactivity of patient sera to epitopes of this region of VZV gE_HP_. Translated into the vaccine situation, this would suggest that gE_HP_ has a greater potential to present the relevant GC-1 epitopes to the immune response than gE_L_, for which such epitopes are blocked by additional O-linked glycans. It is possible that different batches of CHO-K1 cells exhibit different glycosylation status and, therefore, generate differences in site occupancy regarding O-linked glycans, making an extrapolation of our data uncertain. However, it has previously been shown that CHO-K1 cells express only a handful of GalNAc transferases [37] and that externally added GalNAc transferase increases O-glycan site occupancy [38]. This indicates that CHO-K1 cells are, in fact, defective in O-linked glycan initiation, generating less densely glycosylated proteins and, therefore, could be suitable for vaccine production.

In sharp contrast to the function above, O-linked glycans may also be engaged in promoting reactivity of viral glycopeptide epitopes to cognate antibodies from convalescent sera, which has been investigated in detail for MLD-containing glycoproteins of herpes simplex virus type 1 and 2 and Epstein–Barr virus (EBV) [22,23,34]. Model studies demonstrated that the innermost GalNAc unit of an O-glycan-containing peptide epitope constitutes an equally efficient binding determinant to the binding paratope of the cognate antibody as does any of its surrounding amino acids [21]. Elimination of the GalNAc unit results in an immunologically inert peptide epitope with no reactivity to patient sera. This phenomenon is of relevance here for a family of glycopeptide stretches situated in the GC-III cluster within the MLD region of gE_HP_, where each of the same short peptide sequences may form immunologically distinct glycopeptide GalNAc glycoforms, dependent on the exact position of the GalNAc unit attached to the peptide. The glycopeptide E505–P522 is of particular interest in this context, since different patient sera reacted to different glycoforms of this glycopeptide (Figure 6). One interpretation of this phenomenon, which also has been demonstrated for the seroreactivity of EBV-infected individuals [21], is that even for a short peptide sequence, different patients modify different glycosylation sites with GalNAc, owing to patient-dependent differences in expression levels of the 20 human glycosyltransferases in the cells in which the virus replicates.

At present, it is not fully clear as to what extent differences in O-linked glycosylation may account for the difference in vaccine efficacy between Zostavax^®^ and Shingrix^®^, but our data provide detailed information as to how O-linked glycans affect presentation of important B cell epitopes. Firstly, previously published data demonstrate that production of gE_L_ in fibroblasts results in extensive O-linked glycosylation, both with respect to the number of glycans and the size of each glycan compared with the current data for gE_HP_, produced in CHO cells [16]. The results here presented clearly demonstrated that O-linked glycans of the GC-I cluster (Figure 2) interfere with the display of antibody recognition epitopes, targeted by antisera from patient that have recovered from VZV infection. The lower rate of O-glycan site occupancy of gE_HP_ compared with gE_L_ may therefore constitute an advantage for Shingrix^®^ to induce B cell responses to VZV gE that mimic the protective antibody profile induced by an active VZV infection. Secondly, data accumulated here for VZV infection, and elsewhere for EBV infection, indicate that different individuals that have responded to a virus infection induce antibodies to different glycopeptide glycoforms of a defined peptide stretch, even as short as 5–8 amino acids long [21,34]. Our data demonstrated that most of these different glycoforms of the various glycopeptides that reacted to patient sera were in fact represented among the gE_HP_ glycoprotein species produced in CHO cells. Thirdly, we observed that within the total fraction of gE_HP_, only a subpopulation of the peptides carried O-linked glycans at defined positions. Also, it appeared that certain positions within gE_HP_ carried an O-linked glycan at higher frequency than other positions, indicating that the protein is also heterogeneously decorated with O-linked glycans when produced in CHO cells. This suggests that Shingrix^®^ based on gE_HP_ is versatile as an immunogen, with the capacity to induce antibodies to a wide variety of the variable glycopeptide epitopes that may be expressed in an idiotypic manner among patients. The finding of minute amounts of nonhuman sialic acid, NeuGc, in gE_HP_ suggests that this construct may have additional adjuvant qualities in comparison to gE_L_. However, this phenomenon needs further confirmation in a separate study.

The difference in O-glycosylation patterns between Zostavax^®^ and Shingrix^®^ may well be explained by important differences in the principles for their respective productions. Thus, Zostavax^®^ is produced in a diploid human cell line, characterized by a dense O-linked glycosylation [39] and, moreover, the vaccine consists of mature virus particles, two factors favoring a high degree of glycosylation site occupancy with mature O-linked glycans. By contrast, Shingrix^®^ constitutes molecular gE, obtained from a tumor cell line characterized by moderately sized glycans [36]. Moreover, our data suggest that gE production in CHO cells is leaky, permitting different gE glycosylation intermediates into the vaccine preparation. Although the initial step in O-linked glycosylation, i.e., the addition of the innermost GalNAc units to the gE_HP_ MLD could be carried out in a seed and spread manner (i.e., some O-glycosylation sites can only accept GalNAc provided that adjacent O-glycosylation sites are already occupied; see [20]), the elongation of the glycan chain seems less well controlled in CHO cells compared to diploid human fibroblasts. This would enable the biosynthesis of multiple gE_HP_ intermediate glycoforms in the producer cell, which may explain the unique capability to express most of the many variable glycopeptide epitopes targeted by the panel of positive VZV sera analyzed here. 

Evidently, the data of the present study focusing on one biochemical trait cannot explain, alone, the complex biology behind the difference vaccine efficacy between Zostavax^®^ and Shingrix^®^. However, the results here demonstrate important differences between the two regarding the major posttranslational modifications of an immunologically active component of both vaccines. Moreover, all these differences are associated with improved presentation of potential neutralization epitopes to B cell immunity effectors, and may therefore account, at least partly, for the higher reported efficacy of Shingrix^®^ compared with Zostavax^®^, a notion that may be of a broader vaccinology relevance. Thus, in addition to herpesviruses, viral glycoproteins equipped with MLDs are major pathogenetic factors for several severe viral infections, including those caused by Ebola virus, Zika virus, tick-borne encephalitis virus, and others [40,41,42,43,44]. Efforts to optimize the O-linked glycan patterns are likely to form a cornerstone in the design of efficient vaccines, also against these viruses.

## 4. Material and Methods

### 4.1. Expression of Recombinant gE in CHO Cells

The protocol for producing gE_HP_ in CHO-K1 cells has been previously described [31]. Briefly, the coding sequence of the extracellular domain from the VZV strain Dumas (amino acids 1–539) was amplified by PCR using DNA from VZV-infected green monkey kidney (GMK) cells and inserted into a pcDNA6/myc-His A vector (Invitrogen, Carlsbad, CA, USA) in frame with a 3′ 6×His-tag, creating a pcDNA6/VZVgE/His construct. The CKO-K1 subclone of CHO cells was obtained from the American Tissue Type Collection. The CHO-K1 cells were transfected with the pcDNA6/VZVgE/His construct complexed with Lipofectamine 2000 (Invitrogen, Carlsbad, CA, USA) and, thereafter, the positive clones were selected in medium containing 10 µg/mL Blasticidin-HCl (Invitrogen, Carlsbad, CA, USA). The clones selected for high expression of the gE_HP_ were expanded and adapted to serum-free suspension growth. In order to produce significant amounts of gE_HP_, the selected clones were grown in a 3 L bioreactor (Applikon Biotechnology, Schiedam, The Netherlands). Cell-free growth medium was harvested continuously and, in total, 12.5 L gE-containing medium was collected. The collected bioreactor product was concentrated by centrifugation and thereafter applied to a HiTrap chelating column (GE Healthcare, Uppsala, Sweden) loaded with Co^2+^. The bound gE-6×His protein was eluted, and the identity verified by Western blot using a mouse monoclonal antibody VZV g Esc-56994 (Santa Cruz Biotechnology, Santa Cruz, CA, USA) and a goat anti-mouse immunoglobulin antibody coupled to alkaline phosphatase (Southern Biotech, Birmingham, AL, USA). The protein concentration was determined using a BCA protein assay kit (Thermo Scientific, Rockford, IL, USA).

### 4.2. LC–MS/MS Analysis

The gE_HP_ (10 µg) was protease digested with trypsin (sequence grade, Promega) and (3 µg) with pronase (*Streptomyces griseus;* Roche Diagnostics) using a modified filter-aided solid phase (FASP) extraction procedure [45]. For this, 30 kDa MWCO Pall Nanosep (Sigma) was used. Cysteine alkylation was carried out using dithiothreitol (DTT) and *S*-methyl methanethiosulfonate (Thermo Scientific). The digestion buffer, 50 mM TEAB (triethylammonium bicarbonate), was supplemented with 1% SDC (sodium deoxycholate) which was removed with 10% TFA (trifluoric acid) after digestion. Peptides was purified according to the protocol using Pierce C18 spin columns (Thermo Scientific) and dissolved in 15 µL of 0.1% formic acid in 3% acetonitrile.

Glycopeptides were analyzed on an Orbitrap Fusion Tribrid mass spectrometer interfaced to an Easy-nLC 1000 (Thermo Fisher Scientific). Peptides (2 µL injection volume) were separated using an analytical column (200 mm × 0.075 mm I.D., alternatively, 350 mm × 0.075 mm I.D.) packed in-house with 3 μm (20 cm long, MS method 1) or 1.8 μm (35 cm long, MS method 2) Reprosil-Pur C18-AQ particles (Dr. Maisch, Germany). The following gradient was run at 200 nL/min and 55 °C; 3%–50% B-solvent (acetonitrile in 0.2% formic acid) over 30 min, 50%–80% B over 5 min, with a final hold at 80% B for 10 min. Ions were injected into the mass spectrometer under a spray voltage of 1.6 kV in positive ion mode. MS scans were performed at 120,000 resolution, *m/z* range 380–1800, and MS/MS analysis was performed in a data-dependent mode, with a top speed cycle of 3 s for doubly or multiply charged (up to z = 7) precursor ions. There were two sets of MS method parameters. 

Settings for MS method 1: Firstly, the precursors with the highest charged state and then the most intense ions in each MS scan over threshold 50,000 were selected for fragmentation (MS2) by high energy collision-induced dissociation (HCD) at 30%, with detection in the Orbitrap at 30,000 resolution. When product ions at *m/z* 204.0867, 138.0545, or 366.1396 were present, then the fragmentation (MS2) for those precursors were triggered. This second MS2 was performed with electron transfer dissociation (ETD) at 100 ms with 200 K reagent target and supplemental high energy collision-induced dissociation (HCD) of 10%, with detection in the Orbitrap at 60,000 resolution, *m/z* range 120–2000. Precursors were isolated in the quadrupole with a 2.0 *m/z* window, and dynamic exclusion within 10 ppm for 20 s was used for *m/z* values already selected for fragmentation. 

Settings for MS method 2: Sequentially to the MS2 scan HCD at 30% as above, a HCD MS2 scan at 40% was performed. Precursors were here isolated in the quadrupole with a 2.5 *m/z* window.

### 4.3. LC–MS/MS Data Analysis

The MS/MS raw files were transformed into Mascot generic format (.mgf) using Mascot distiller (Matrix Science, London, UK) and sequence searches were conducted using an in-house Mascot server. The search database consisted of the gE sequence (Uniprot ID P09259), and the enzyme was set to *trypsin* or to *no enzyme* for the trypsin- and pronase-cleaved samples, respectively. Methylthio-derivatization of Cys (45.9877 u) was set as a static modification, and Met oxidation was set as an allowed modification. For HCD spectral searches, HexNAc (203.0794 u), HexHexNAc (365. 1322 u), NeuAcHexHexNAc (656.2276 u), and (NeuAc)_2_HexHexNAc (947.3230 u) were set as allowed modification of Ser, Thr, and Tyr residues and, additionally, the same masses were set as allowed neutral losses of peptide backbone fragmented ions containing the arbitrarily chosen glycosylation site. For ETD spectral searches, the same glycan modifications were used but without any fragmentation loss of the glycans. For pronase samples, extracted ion chromatograms of diagnostic oxonium ions were prepared, for instance, *m/z* 138.055 for HexNAc-containing glycopeptides and *m/z* 290.087 for NeuGc-containing glycopeptides. Hits were assigned by using the tentatively assigned peptide ion, or the peptide+HexNAc ion for N-glycopeptides, as input in the Find peptide web service (Expasy, Swiss institute of bioinformatics). The glycan structure for each glycopeptide was assigned by annotating the glycosidic fragmentation pattern together with the mass difference between the precursor and MS/MS-generated peptide ions.

### 4.4. Deglycosylation of gE_HP_

Neuraminidase (α2-3, α2-6, and α2-8 linked sialic acid residues), PNGaseF (high mannose, hybrid and complex type N-linked glycans), and O-glycosidase (core 1 and core 3 O-linked glycans) (all from New England Biolabs, Ipswich, MA, USA) were used to remove glycans from the gE_HP_. In brief, 40 µg gE_HP_ was combined with 10× glycobuffer to a volume of 20 µL. Subsequently, either 4 µL neuraminidase (5 × 10^4^ U/mL), 4 µL PNGaseF (5 × 10^5^ U/mL), or a combination of 4 µL neuraminidase and 4 µL O-glycosidase (4 × 10^7^ U/mL) was added, together with ddH_2_O, to achieve a final volume of 40 µL. The samples were subsequently incubated at 37 °C for one hour and then placed on ice before storage at −20 °C. To evaluate the glycan removal, the glycosidase-treated gE_HP_ was incubated with sample buffer and heated at 95 °C for 10 min. The samples were loaded on a Novex NuPAGE Bis-Tris 4%–12% 1 mm gel using MOPS running buffer including NuPAGE antioxidant prepared according to the manufacturer instructions (Invitrogen, Carlsbad, CA, USA). For protein size determination, the PageRuler Prestained Protein Ladder (Thermo Scientific, Rockford, IL, USA) was included in the gel. The gel was subsequently stained with Novex Colloidal Blue stain according to the instructions from the manufacturer (Invitrogen, Carlsbad, CA, USA). 

### 4.5. Serology Using Deglycosylated gE_HP_

The different preparations of partially deglycosylated gE_HP_ were diluted to a final concentration of 1 ng/mL in carbonate buffer pH 9.6, and 100 µL of each diluted preparation was applied to a 96-well Maxisorp-plate. Additionally, untreated gE_HP_ at a concentration of 1 µg/mL was added as a control. The plate was sealed with an adhesive plastic film and incubated at 4 °C, overnight. After adsorption of the antigens, the plates were washed three times in wash buffer (PBS supplemented with 0.05% Tween 20). To prevent unspecific binding, the plates were incubated with blocking solution (PBS with 2% skimmed milk) for 30 min at room temperature. Thereafter, the plates were washed three times in wash buffer before addition of human sera from VZV-positive individuals, with high anti-VZV IgG antibody titer (n = 5) and intermediate anti-VZV IgG titer (n = 5) as determined by ELISA (whole virus or gE antigen) and immunofluorescence of VZV-infected cells. All sera were obtained from the routine diagnostic laboratory of Clinical Virology, at the Sahlgrenska University Hospital, Gothenburg, Sweden. The Medical Ethics Committee at University of Gothenburg approved the study; Dnr: 307-06. The study has been performed in accordance with the ethical standards laid down the Helsinki agreements and its later amendments. The serum samples were diluted in PBS containing 1% skimmed milk and 0.05% Tween 20. The plates were incubated at 37 °C for 90 min. After an additional three washes in wash buffer, 100 µL alkaline phosphatase-conjugated goat anti-human IgG antibody, diluted 1:1000 in PBS with 1% skimmed milk and 0.05% Tween 20, was added to each well. The plates were then incubated at 37 °C for one hour and thereafter washed three times in wash buffer. Finally, 200 µL of 1 mg/mL phosphatase substrate (4-nitrophenyl phosphate disodium salt hexahydrate) (Sigma Aldrich, St. Louis, MO, USA), diluted in diethanolamine (DEA) buffer, was added to each well, and absorbance was measured after 20 min at 405 nm with background reading at 620 nm using an E_max_ Precision microplate reader (Molecular Devices, Sunnyvale, CA, USA). 

### 4.6. General Procedure for Synthesis of Peptide Library

Peptides were prepared by Fmoc solid-phase peptide synthesis on a Syro Wave automated peptide synthesizer (Biotage, Sweden). Syntheses were carried out on TentaGel R Rink Amide 0.19 mmol/g, (Rapp Polymers GmbH, Germany). Amino acids had Fmoc protection of Nα-amino groups; side chain-protecting groups were *tert*-butyl (Ser, Thr, Glu, and Asp), 2,2,4,6,7-pentamethyldihydrobenzofuran-5-sulfonyl (Pbf, for Arg), trityl (Trt, for Asn, Gln, and His), and *tert*-butyloxycarbonyl (Boc, for Lys); acetonitrile, formic acid, triethylsilane (TES), sodium methoxide solution (0.5 M), trifluoroacetic acid (TFA), and dichloromethane (DCM) were purchased from Sigma-Aldrich (Denmark). Common amino acids, 8-(9-fluorenylmethyloxycarbonyl-amino)-3,6-dioxaoctanoic acid (Fmoc-O2Oc-OH), and other reagents for peptide synthesis were supplied by Iris Biotech (Germany). *N*-α-Fmoc-*O*-β-(2-acetamido-2-deoxy-3,4,6-tri-*O*-acetyl-β-D-galactopyranosyl)-L-threonine (Fmoc-Thr(β-D-GalNAc (Ac)3)-OH) was supplied by Sussex Research Laboratories Inc. (Ontario, Canada). Purification was performed by RP-HPLC (Dionex Ultimate 3000 system) with preparative C4 (Phenomenex, Luna 300Å 5 µm C4 particles, 21.5 × 250 mm and C18 columns FeF Chemicals, 300Å 5 µm C18 particles, 21.5 × 250 mm) using the following solvent system: water containing 0.1% TFA (solvent A), and methanol containing 0.1% TFA (solvent B). 

Synthesis was carried out on a TentaGel R Rink Amide 0.19 mmol/g using 5 equivalents of amino acids and 1-hydroxy-7-azabenzotriazole (HOAt), 4.8 equivalents of *N*-[(1*H*-benzotriazole-1-yl)(dimethylamino)methylene]-*N*-methylmethanaminium hexafluorophosphate *N*-oxide (HBTU) and 9.8 equivalents *N*,*N*-diisopropylethylamine (DIEA). During coupling of Fmoc-Thr(β-D-GalNAc(Ac)3)-OH only 1.1 equivalent of the amino acid and HOAt, 1 equivalent HBTU, and 1.7 equivalent of DIEA was used. Double couplings were performed throughout the synthesis with a coupling time of 2 × 10 min at 75 °C with NMP, and washing in between. For the first 20 couplings, deprotection of Nα-Fmoc was done by addition of piperidine–DMF (2:3) for 3 min, followed by piperidine–DMF (1:4) for 2 × 10 min, then a washing step with NMP and further piperidine–DMF (1:4) for 10 min. After each coupling and deprotection, a washing procedure with 3 × NMP, 1 × DCM, and 3 × NMP was performed. From then on, two extra 10 min deprotection steps with piperidine–DMF (1:4) were performed. The resin was washed six times with DCM after completion of synthesis.

The peptides were released from the solid support by treatment with trifluoroacetic acid (TFA), triethylsilane (TES), and H_2_O (95:2:3) for 2 h. The TFA solutions were concentrated by nitrogen flow, and the compounds were precipitated with diethyl ether to yield the crude product. The crude peptide was purified on a C4 column, followed by three subsequent purification steps on a C18 column. Peptide identification was carried out by electrospray ionization mass spectrometry (ESI–MS) (MSQ Plus Mass Spectrometer, Thermo). Purity was evaluated by analytical HPLC (Dionex Ultimate 3000 system) on an analytical C18 column (Phenomenex Gemini NX 110 Å, 5 µm, C18 particles, 4.60 × 50 mm) using a linear gradient flow of water–methanol containing 0.1% formic acid (5%–100% methanol for 15 min). 

### 4.7. Print of Microarrays

All peptides were diluted 1:20 in print buffer (150 mM sodium phosphate, 0.005% CHAPS{3-[(3-cholamidopropyl)-dimethylammonio]-1-propanesulfonate}, 0.03% NaN_3_; pH 8.5). Peptides 27–48 were further diluted to 1:60. The peptides were then immobilized on *N*-hydroxysuccinimide (NHS) slides (Schott Nexterion, SlideH) on a 16-well format with 18 × 18 spots, with a Biorobotics Microgrid II spotter (Genomics Solution) using Stealth 3B microspotting pins (Arrayit) with approximately 6 nL per spot. Each peptide was deposited in triplicates in 16 identical subarrays. After printing, the slides were incubated at 70% humidity for 60 min, and the remaining NHS groups were deactivated in blocking buffer (50 mM ethanolamine in 50 mM borate buffer; pH 8.5) for 2 h, and then rinsed in Millipore water and spun dry.

### 4.8. On-Slide Glycosylation of Synthetic Peptides

Blocked slides were fitted with a 2-well superstructure (Fast frame, Schleicher & Schuell (Whatman)), to form two wells. The wells were filled with 500 µL of glycosylation mixture (10 µL of 100 mM UDP-GalNAc, 10 µL of either 0.36 mg/mL GalNAc-transferase 2 (GalNAc-T2), or 0.46 mg/mL GalNAc-transferase 3 (GalNAc-T3) 10 mM, and 480 µL HEPES buffer (10 mM containing 0.1 mM CaCl_2_, 2 mM MnCl_2_, 0.15 mM NaCl; pH 7.5), placed in a humidification chamber, and incubated for 2 h at RT. Slides were then washed with 0.1 M acetic acid (2 × 5 min, shaking) and PLI-P (5 min, shaking). Slides were again washed with PBS, rinsed thoroughly with water, dried by centrifugation, and were then ready to be immediately used in a subsequent lectin-binding experiment. Slides were incubated with biotinylated Vicia villosa (VVA) lectin (1:500 dilution in PLI-P buffer (0.5 M NaCl, 3 mM KCl, 1.5 mM KH_2_PO_4_, 6.5 mM Na_2_HPO_4_, and 3% bovine serum albumin (BSA); pH = 7.4) for one hour at room temperature, followed by incubation with streptavidin-AlexaFluor 647 (1:1000 dilution in PLI-P) for one hour. All incubation steps were performed in a humidification chamber and were separated by three washing steps in phosphate buffer saline (PBS). After a final wash in PBS, slides were rinsed in water, dried by centrifugation, and scanned. The images were analyzed, quantified, and scanned as described below.

### 4.9. Serology Using a Synthetic Peptide Epitope Microarray

Sera either positive (n = 11) for VZV IgG or negative for VZV IgG (n = 11), as determined by ELISA (whole virus and gE antigen) and immunofluorescence of VZV-infected cells, were diluted (1:5) in PLI-P buffer (0.5 M NaCl, 3 mM KCl, 1.5 mM KH_2_PO_4_, 6.5 mM Na_2_HPO_4_, and 3% BSA; pH = 7.4). All sera were obtained from the routine diagnostic laboratory of Clinical Virology, at the Sahlgrenska University Hospital, Gothenburg, Sweden. For the secondary staining, goat anti-human IgG-Cy3 construct (10 μg/mL; Sigma-Aldrich) was diluted in PLI-P buffer 1:500. The subarray wells were filled with 90 μL each of sample solution (primary or secondary alike). All incubation steps were performed in a humidification chamber at room temperature (overnight for sera incubation and one hour for secondary staining) and were separated by three wash steps in PBS. After a final wash in PBS, slides were rinsed in water, dried by centrifugation, and scanned. The images were analyzed and quantified. The slides were scanned with a microarray scanner (ProScanArray; Perkin Elmer) equipped with three lasers for excitation at 488, 543, or 633 nm. The scanned images were analyzed with Scanarray Express software. For Cy3 fluorescence, 543 nm (excitation) and 570 nm (emission) were used. Spots were identified using automated spot-finding with manual adjustments for occasional irregularities. The mean value of relative fluorescence intensity was used, and the spot intensities were determined by subtracting the median pixel intensity of the local background from the average pixel intensity within the spots. Triplicate spots were averaged. Serum samples with a relative fluorescence value higher than two standard deviations over the mean of the control group were designated reactive towards a peptide.

### 4.10. Statistical Analysis

Average relative fluorescence (RFU) values were compared using one-way ANOVA and Tukeys´s multiple comparisons test using the GraphPad 6 Prism software (GraphPad Software Inc., San Diego, CA, USA). A *p*-value < 0.05 was considered significant.

## 5. Conclusions

We defined that the glycosylation profile of VZV gE expressed in CHO-K1 cells, the same cell type used for production of the subunit vaccine Shingrix, is different compared to gE derived from human fibroblasts, that are used for production of the live vaccine Zostavax. The O-linked glycan occupancy is less dense in gE from CHO-K1 cells and this may increase the exposure of the B cell epitopes compared to the situation in gE from fibroblasts. We also found evidence for both interference and enhancement of antibody binding, that were dependent on specific O-linked glycans. The results demonstrate important differences between Shingrix and Zostavax and could be part of an explanation for the observed superior efficacy of the subunit vaccine.

## Figures and Tables

**Figure 1 ijms-20-00954-f001:**
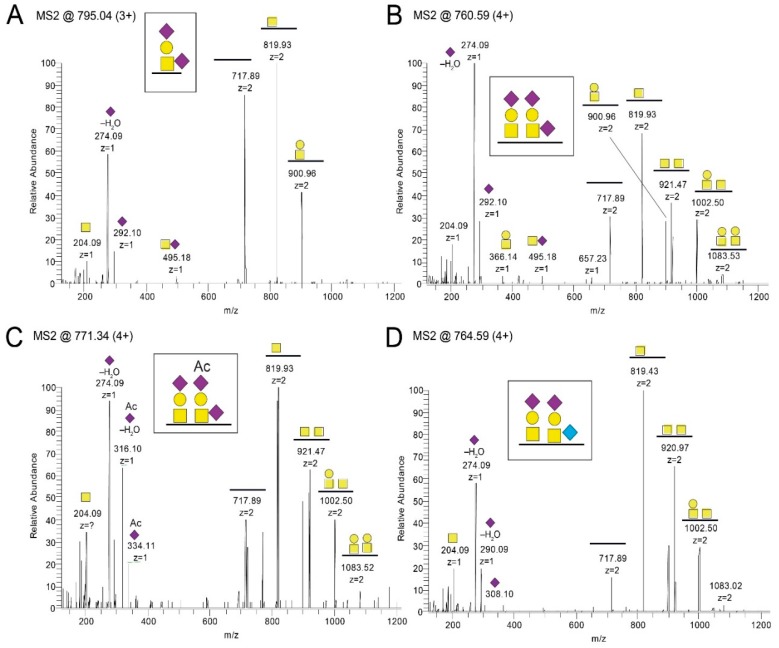
LC–MS/MS spectra after pronase and trypsin treatment of gE_HP_. Shown are representative spectra covering amino acid 325-TRNPTPAVTPQPR-337. T329 and T333 were found to (**A**,**B**) carry monosialylated and disialylated core 1 O-linked glycans. Also, (**C**) NeuAc(Ac) variants and (**D**) NeuGc variants were identified. In total, trypsin and pronase treatment resulted in 62% coverage of the total protein. 
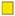

*N*-acetylgalactosamine (GalNAc), 
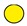
 galactose (Gal), 
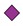

*N*-acetylneuraminic acid (NeuAc), and 
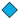

*N*-glycolylneuraminic acid (NeuGc).

**Figure 2 ijms-20-00954-f002:**
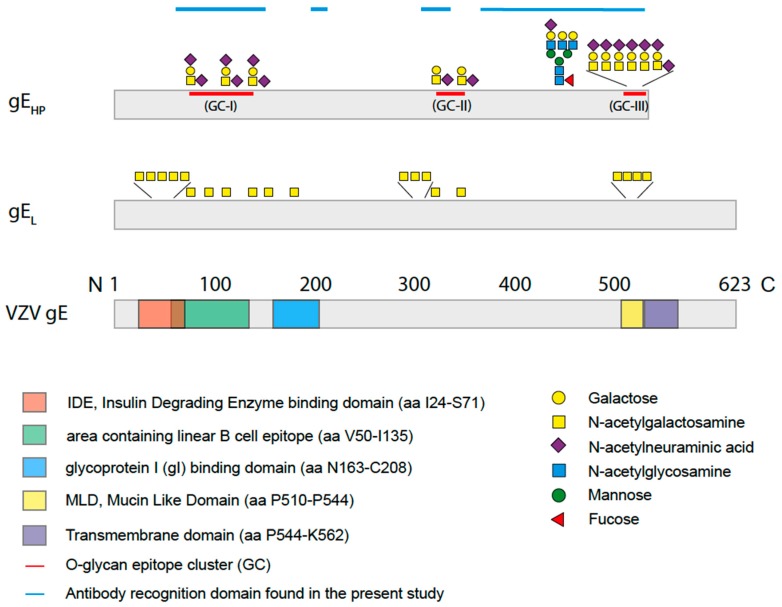
Schematic drawing of varicella zoster virus (VZV) gE, gE_L_, and gE_HP_ with an outline of the glycan compositions and positions on the peptide sequence of gE_HP_ as determined by mass spectrometry in the present study. Three O-linked glycan epitope clusters (GC) of sialylated core 1 O-linked glycans and one complex type N-linked glycan were found. Two O-linked glycans were situated within the linear B cell epitope (green) [29] while the largest cluster of O-linked glycans was found in the mucin-like domain (MLD) (yellow) close to the transmembrane domain (purple). For gE_HP,_ neither the IDE domain (red) nor the gI-binding domain (blue) contained any glycans, while O-linked glycans were found in the corresponding domains of gE_L_ [30].

**Figure 3 ijms-20-00954-f003:**
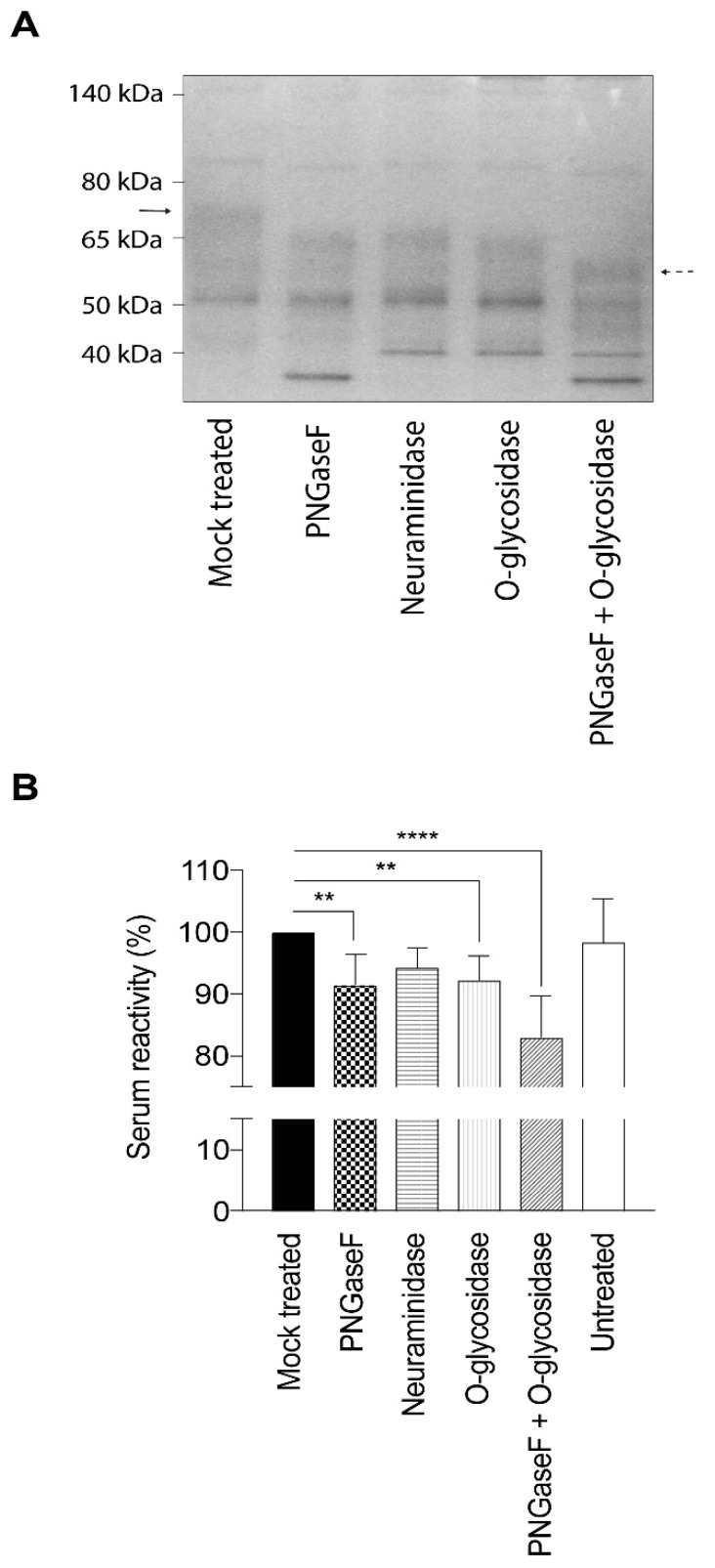
Enzymatic deglycosylation of gE_HP_ reduces the reactivity towards serum from individuals earlier infected with VZV. Glycans were removed by PNGaseF, neuraminidase, and O-glycosidase treatment, respectively. (**A**) Deglycosylation of gE_HP_ was verified by gel electrophoresis and the enzymatic hydrolysis resulted in distinct size shifts. Solid arrow indicates fully glycosylated gE_HP_ and dashed arrow indicates gE_HP_ stripped of both N- and O-linked glycans. (**B**) Serum reactivity towards gE_HP_, either unmodified or deglycosylated, was assessed by ELISA using serum from VZV+ individuals (n = 10). Statistic calculations were done using one-way ANOVA with Tukey’s multiple comparisons test (significance is indicated by ** *p*-values < 0.01 and **** *p*-values < 0.0001).

**Figure 4 ijms-20-00954-f004:**
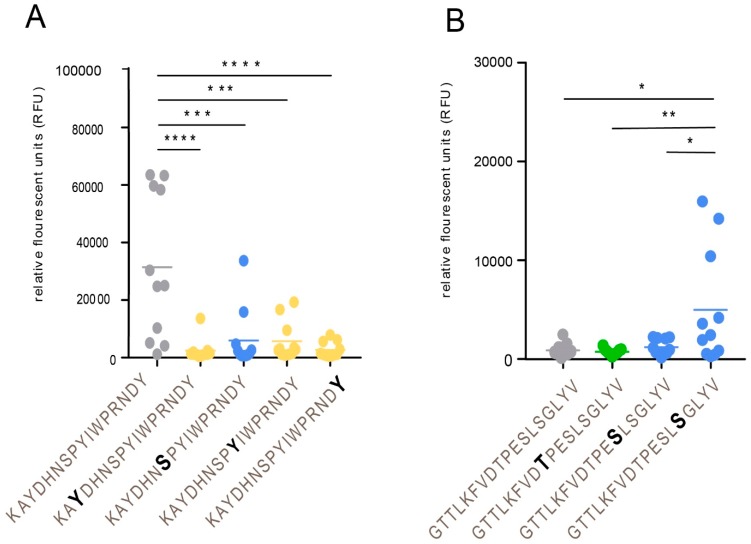
Addition of a GalNAc residue to a peptide epitope: effects on antibody binding. (**A**) Interference with antibody binding. (**B**) Enhancement of antibody binding. Individual serum samples from VZV-positive patients (n = 11) were analyzed. Addition of a GalNAc residue is indicated by bold letters. Statistic calculations were done using one-way ANOVA with Tukey´s multiple comparisons test (significance is indicated by * *p*-values < 0.05, ** *p*-values < 0.01, *** *p*-values < 0.001, and **** *p*-values < 0.0001).

**Figure 5 ijms-20-00954-f005:**
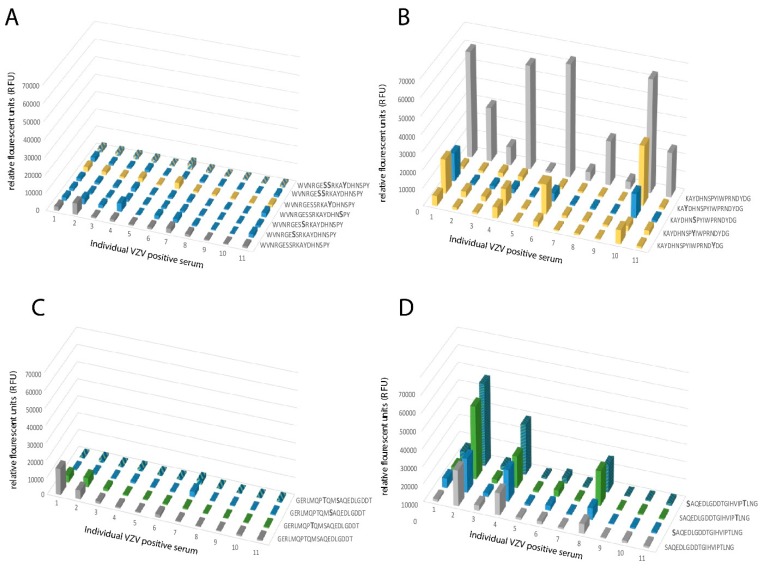
A homogenous response towards synthetic peptides with single or multiple GalNAc modifications was observed for peptides covering parts of the N-terminal region of gE. (**A**) Synthetic peptide 64-WVNRGESSRKAYDHNSPY-81 unmodified or modified by GalNAc residues at either Ser or Tyr, or a combination of both. (**B**) Synthetic peptide 73-KAYDHNSPYIWPRNDYDG-90 unmodified or modified by GalNAc residues at either Ser or Tyr. (**C**) Synthetic peptide 111-GERLMQPTQMSAQEDLGDDT-130 unmodified or modified by GalNAc residues at either Ser or Thr, or a combination of both. (**D**) Synthetic peptide 121-SAQEDLGDDTGIHVIPTLNG-130 unmodified or modified by GalNAc residues at either Ser or Thr. All peptides were incubated with serum samples and the relative fluorescence unit (RFU)-values were determined. Grey: unglycosylated peptide. Blue: GalNAc modification on a Ser residue. Yellow: GalNAc modification on a Tyr residue. Green: GalNAc modification on a Thr residue. Blue and yellow: GalNAc modification on Ser and Tyr residues. Blue and green: GalNAc modification on Ser and Thr residues. X-axis shows the individual serum samples (n = 11) from VZV IgG-positive patients, plotted in the same order of individual sera in all graphs.

**Figure 6 ijms-20-00954-f006:**
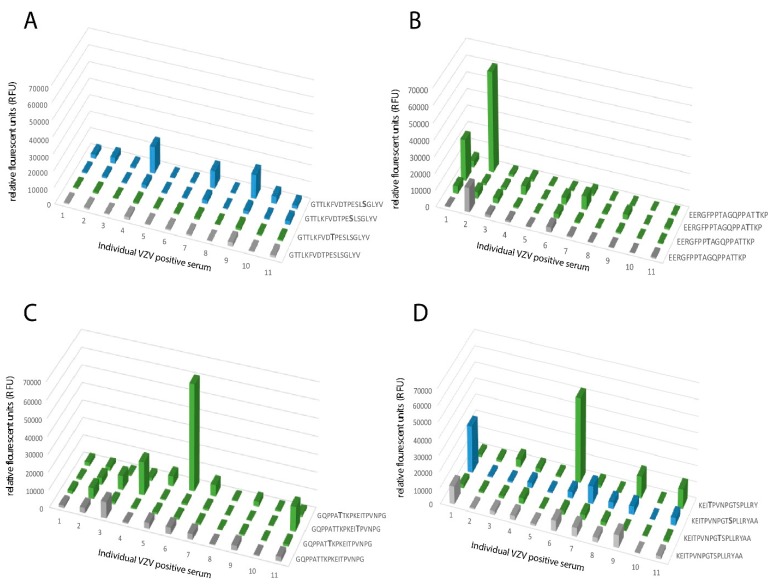
A heterogenous response towards synthetic peptides, with single or multiple GalNAc modifications, was observed for peptides covering parts of gE close to the transmembrane domain, including the MLD domain when individual serum samples were analyzed in detail. (**A**) Synthetic peptide 460-GTTLKFVDTPESLSGLYV-477 unmodified or modified by GalNAc residues at either Ser or Thr. (**B**) Synthetic peptide 505-EERGFPPTAGQPPATTKP-522 unmodified or modified by GalNAc residues at Thr. (**C**) Synthetic peptide 514-GQPPATTKPKEITPVNPG-531 unmodified or modified by GalNAc residues at Thr residues. (**D**) Synthetic peptide 523-KEITPVNPGTSPLLRY-540 unmodified or modified by GalNAc at either Ser or Thr residues. Grey: unglycosylated peptide. Blue: GalNAc modification on a Ser residue. Green: GalNAc modification on a Thr residue. X-axis shows the individual serum samples (n = 11) from VZV IgG-positive patients, plotted in the same order of individual sera in all graphs.

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
