# Peer review of "Recombinant Glycoprotein E of Varicella Zoster Virus Contains Glycan-Peptide Motifs That Modulate B Cell Epitopes into Discrete Immunological Signatures"

_ijms, 2019, doi:10.3390/ijms20040954_

Reviewer 1 Report

The subject is according to the scope of the Journal. I appreciated the deep lab work. The manuscript is well written, logical and intelligible.

Author Response

We have edited the manuscript to correct for minor language mistakes. No further comments are required.

Reviewer 2 Report

In their manuscript “Recombinant glycoprotein E of varicella zoster virus contains glycan-peptide motifs that modulate B cell epitopes into discrete immunological signatures”, Nordén et al; describe experiments aimed at characterization of the glycosylation status of varicella zoster virus (VZV) glycoprotein E (gE). This study was prompted by the observation that a recently licensed recombinant subunit vaccine (Shingrix), based on recombinant gE produced in CHO-K1 cells, produces stronger virus-specific CD4+ T cell and antibody B cell responses to gE compared to those promoted by the currently used live attenuated vaccine (Zostavax). Zostavax is produced in human fibroblasts, so the authors hypothesize that gE glycosylation might be different in CHO-K1 cells, and that this may underlie the improved profile of the recombinant gE-based Shingrix vaccine.

First, using mass spectrometry, the authors show that O-glycan site occupancy is reduced in gE generated in CHO-K1 cells when compared to the previously reported site occupancy in gE derived from human fibroblasts. Second, using peptide/glycopeptide libraries to scan immunodominant epitopes they go on to show that this glycosylation profile promoted increased binding of IgG antibodies from serum derived from VZV seropositive individuals because of a decreased number of masking glycans. Interestingly, the authors also show that in some cases glycans actually enhance IgG binding. Accordingly, these data suggest a mechanistic basis for the superior characteristics of the Shingrix vaccine. However, as the authors point out, vaccine challenge initiates a train of complex processes, and other factors are surely involved in the increased efficacy of the CHO-K1-derived gE based vaccine. Nevertheless, the data highlight the important principal that varying glycan site occupancy during generation of protein-based vaccines may be worth looking at when optimizing vaccine efficiency. 

Major points

1)   Different glycosylation profiles are seen in proteins expressed in human fibroblasts and CHO-K1 cells. But equally, different glycosylation profiles can be seen in proteins derived from the same cell line grown under different culture conditions. So, what is the relationship between the CHO K1-expressed gE reported here and that found in the Shingrix gE? In other words, how certain are the authors that the glycosylation status of their CHO-K1 expressed protein is the same as that of the Shingrix protein?

Minor points

1)   Figures: although the authors use the standard symbols for sugars, not all readers will understand their meaning without having to search for the information. So, the symbols should be defined in the legends to the figures in which they first appear.

2)   Figure 2 shows an outline of the glycans/glycan occupancy position found in the present study. I understand that it might be difficult to represent in a digestible form, but it would help if the reader could compare visually this data with that obtained for gE derived from human fibroblasts (Zostavax situation).

3)   Methods: where was the pronase purchased?

4)   What is the significance of the small amounts of the non-human N-glycolylneuraminic acid on O-glycans in the GC-I and -II clusters? Does the Shingrix product contain such structures, or are the CHO cells used by the vaccine manufacturer modified in order to block NeuGc production (humanized)?

5)   In figure 3A (and text lines 191-193), the dashed arrow indicates the migration position of “fully stripped” gE, but is this the migration position of the non-glycosylated bacterially expressed gE?. Does the neuraminidase used remove NeuGc residues? I assume that some of the bands that we see correspond to the glycosidases used, but even in the mock digestion lane there seem to be many bands, some of which appear to respond to the different enzyme treatments in similar fashion to gE. Could the presence of these non-gE bands contribute to the signals seen in 3B?

6)   Discussion: lines 399 – 402. Explain “seed and spread”

Author Response

Major points

1. The referee is right in pointing out the sensitivity of the glycosylation process to various stress factors, resulting in differences in glycan structures also in the very same cell line. However, this phenomenon is most evident for the peripheral glycosylation steps of viral glycoproteins. But glycosylation site occupancy, at least when it comes to N-linked glycans, is conservative and constant. Therefore, practically identical N-glycosylation sites are occupied, irrespective of tissue origin or stress status (virus infection etc). Regarding O-linked glycans, we found that in human fibroblasts infected with herpesvirus the levels of translated glycosyltransferases were constant over time (Ref 20 in the revised manuscript).  The CHO cells used are of the standard lineage, applied not only for vaccine production, but also for expression of a multitude of different mammalian glycoproteins for medical/nonmedical use. However, it was not possible to perform detailed characterization of gE from the Shingrix formulation as we did not have access to the vaccine during this work. 

Nevertheless, experimental data indicate that there are differences in O-glycan occupancy between gE produced in CHO-cells and fibroblasts, respectively. It has since long been established that CHO cells, like other transformed cell lines, are defective in elongation of O-glycans, resulting in expression of Tn and T antigen on the cell surface (ref 36 in the revised manuscript). To our knowledge, it is not known to what extent O-glycan site occupancy varies between different batches of CHO cell lines. However, it has been shown that the GalNAc-transferases, responsible for initiation of O-linked glycosylation, are only modestly expressed in CHO-K1 cells. Specifically, only four (GalNacT-2, -7, -11 and -19) out of 18 were expressed, strengthening our hypothesis that proteins are less occupied by O-linked glycans in this cell type (ref 37 in the revised version of the manuscript). Moreover, it has also been shown that supplementing CHO-K1 cells with externally added GalNAcT-4 increases O-glycan occupancy of a reporter construct (ref 38 in the revised manuscript).  We thank the referee for pointing out this important issue and we have added a passage and both of the above-mentioned references in the discussion section of the revised manuscript, page 19-20 line 390-396.

Minor points

1. The sugar symbols have been added to figure 1 and figure 2 in the revised manuscript.

2. Figure 2 have been changed accordingly in the revised manuscript.  This addition, thanks to the suggestion from the reviewer, has improved understanding of the comparison between the two glycosylation patterns.

3. The requested information have been added to the method section.

4. This is an important point brought up by the referee. To the best of our knowledge the CHO cells used for Shingrix production has not been knocked out for the gene CMAH, encoding for CMP-N-acetylneuraminic acid hydroxylase that is responsible for synthesizing CMP-Neu5Gc from CMP-Neu5Ac. According to our opinion, this is a factor whose possible effect on Shingrix, owing to its non-self character in the human setting, would be immunity-enhancing rather than immunity-suppressing. Owing to the very small amounts found, we refrained to “advertise” this in the manuscript as a factor further supporting the superior immunogenicity of Shingrix, simply because the aim of our manuscript was not to perform a systematic study of this aspect. But the referee is completely right, the Neu5Gc is a structural component of gE from CHO, and its possible relevance should be commented on, and this is done on page 21, line 434-436 of the revised manuscript.

5. The band corresponding to fully glycosylated gE produced in CHO-K1 cells is indicated by the solid arrow and the stripped gE also from CHO-K1 cells is indicated by dashed arrow as the referee duly notes. Also, there are several bands, in the treated samples, of a size around 40 kDa which corresponds to the added glycosidases as determined by Coomassie Blue staining in Fig 3A. In the mock treated sample and the glycosidase treated samples, one band is prominent at around 52 kDa which corresponds to unglycosylated gE (pgE) regularly reactive in immunoblot with human VZV-positive sera. Presumably, this un-glycosylated pgE will contribute to the reactivity to the gE specific antibodies, seen in figure 3B. Any residual enzymes or non-gE material will most likely not contribute to the specific IgG gE reactivity seen in Fig 3B. However, we thank the referee for highlighting this figure as we noted that we overinterpreted the extent of deglycosylation after enzymatic cleavage. We have changed this passage in the result section, page 10 line 217.

6. We have added an explanatory line regarding the seed and spread concept in the revised manuscript, page 21 line 446-447.